# Why do women attend antenatal care but give birth at home? a qualitative study in a rural Ghanaian District

**Kennedy A. Alatinga**[1]*, **Jennifer Affah**[2], **Gilbert Abotisem Abiiro**[3,4]

1 Department of Community Development, Faculty of Planning and Land Management, SD-Dombo University of Business and Integrated Development Studies, Wa, Ghana, 2 Department of Social Studies, Wa Technical Institute, Wa, Ghana, 3 Department of Population and Reproductive Health, School of Public Health, University for Development Studies, Tamale, Ghana, 4 Department of Health Services, Policy, Planning, Management and Economics, School of Public Health, University for Development Studies, Tamale, Ghana

* kalatinga@gmail.com

## Abstract

### Background

The Sustainable Development Goal Three has prioritised reducing maternal, under-5 and neonatal mortalities as core global health policy objectives. The place, where expectant mothers choose to deliver their babies has a direct effect on maternal health outcomes. In sub-Saharan Africa, existing literature has shown that some women attend **antenatal** care during pregnancy but choose to deliver their babies at home. Using the Andersen and Newman Behavioural Model, this study explored the institutional and socio-cultural factors motivating women to deliver at home after attending antenatal care.

### Methods

A qualitative, exploratory, cross-sectional design was deployed. Data were collected from a purposive sample of 23 women, who attended **antenatal** care during pregnancy but delivered their babies at home, 10 health workers and 17 other community-level stakeholders. The data were collected through semi-structured interviews, which were audio-recorded, transcribed and thematically analysed.

### Results

In line with the Andersen and Newman Model, the study discovered that traditional and religious belief systems about marital fidelity and the role of the gods in childbirth, myths about consequences of facility-based delivery, illiteracy, and weak women's autonomy in healthcare decision-making, predisposed women to home delivery. Home delivery was also enabled by inadequate midwives at health facilities, the unfriendly attitude of health workers, hidden charges for facility-based delivery, and long distances to healthcare facilities. The fear of caesarean section, also created the need for women who attended antenatal care to deliver at home.

**Data Availability Statement:** This study is based on a dataset of 50 qualitative interview transcripts. However, we did not seek ethical permission from the participants, nor the ethics committee to use data for anything else other than for the specific

purposes of this study. For this reason, we do not have the explicit permission for data sharing, re-analysis nor future studies. It would therefore, be inappropriate and unethical to make them available in the public domain. Furthermore, data cannot be shared publicly because the individual transcripts contain very sensitive and identifying personal information from the participants and we did not obtain consent from the participants nor the ethics committee to upload such information for public sharing. Thus, by making the transcripts publicly available, for which consent was not obtained from the participants and ethics committee, will only lead to ethical violations. Even so, qualified individuals can direct queries by contacting Miss Joana Nyamekye Afrifa (irb@navrongo.mimcom.net).

**Funding:** The authors received no specific funding for this work.

**Competing interests:** The authors have declared that no competing interests exist.

## Conclusion

The study has established that socio-cultural and institutional level factors influenced women's decisions to deliver at home. We recommend a general improvement in the service delivery capacity of health facilities, and the implementation of collaborative educational and women empowerment programmes by stakeholders, to strengthen women's autonomy and reshape existing traditional and religious beliefs facilitating home delivery.

## Introduction

Globally, the World Health Organization (WHO) [1] estimated that about 295 000 women died due to causes related to pregnancy and childbirth in 2017. Perhaps, two reasons account for this state of affairs—low skilled delivery and poor quality of antenatal care [2]. For these reasons, the Sustainable Development Goal (SDG) Three has prioritised reducing maternal, under-5 and neonatal mortalities as core global health policy objectives [3]. Place of delivery—the place where expectant mothers choose to deliver their babies has a direct effect on maternal health outcomes. The literature suggests that health facility-based or institutional delivery, manned by trained personnel has the tendency of reducing maternal and new-born mortalities and morbidities than home-based delivery [2]. Studies have shown that 50% of women in the world give birth at home without any professional supervision [1]. Scholarship from sub-Saharan Africa, reported that 60% of mothers do not have a skilled birth attendant (SBA) present during childbirth, and that out of 95% of pregnant women who attended antenatal care (ANC) in health facilities, almost half (47%) of them delivered at home [4].

The Government of Ghana has been promoting institutional delivery through a number of interventions, notably the Community-based Health Planning and Services (CHPS) and the Free Maternal Healthcare Policy (FMHP), under Ghana's National Health Insurance Scheme (NHIS). The CHPS is a community-led participatory planning approach for delivering basic preventive and curative healthcare services to deprived communities, by stationing nurses called Community Health Officers (CHOs) in communities. The core aim of CHPS is to bridge geographical barriers in access to healthcare, by bringing healthcare to the doorsteps of communities. The FMHP sought to address financial barriers in accessing maternal healthcare because persons in need of antenatal, delivery and post-natal healthcare services are entitled to these services, free of charge [5].

Despite these interventions by Government, findings from the 2017 Ghana Maternal Health Survey revealed that 79% of births were delivered at a health facility conducted by professionals, of which 90% occurred in urban areas, compared to 68% in rural areas [6]. In Northern Ghana (including Upper West Region), health professionals attended to only 37% of births [7]. Ghana Health Service reported that more than 95% of pregnant women in the Upper West Region attended at least one ANC visit [8] and approximately 80% attended the recommended four (4) or more ANC visits but less than 60% of them delivered at a health facility assisted by a SBA [8]. Ghana Health Service [9] reported that infant mortality in the Upper West Region stood at 64 per 1,000 live births, while under-5 mortality stood at 92 per 1,000 live births, against the national averages of 41 and 60 per 1,000 live births in 2017 respectively [9]. Particularly in the Jirapa Municipality of the Upper West Region, evidence suggests that during the period 2012–2017, among 14,314 women who attended four (4) or more ANC visits, only 5,351 (37.4%) delivered at the health facility, and the remaining 8,963 (62.6%) delivered at home [9]. At these numbers of institutional deliveries, the Municipality did not meet

the national target of 60% institutional deliveries [10]. The Jirapa Municipal Health Director-ate reports that one major challenge to improving skilled delivery is poor geographical access and some unfavourable cultural norms and traditions [10].

This trend ignites great research interest, to unravel the factors that contribute to women delivering at home, despite attending the recommended ANC. Even so, there is relatively little literature on the reasons why women in Sub-Saharan Africa still deliver at home. Previous studies established that household level factors such as husband's occupation, and the financial status of families, and individual level factors such as age and educational level of women, greatly influence their choices of a place of delivery [2, 11]. Interestingly, most of these studies concentrated on quantifying the number of women who delivered at health facilities, without going beyond these numbers to interrogate and understand the institutional and socio-cultural factors that influenced the women's choice. Understanding these factors is crucial to identify-ing gaps in the existing research, designing appropriate interventions, and developing effective policies for addressing low facility-based delivery rates [12]. It will also help improve maternal health outcomes and contribute to achieving the health-related SDGs. The purpose of this study is to explore the institutional and socio-cultural factors that influence women's decision to deliver at home after attending ANC.

## Theoretical model of the study

This paper draws on the Andersen and Newman's Behavioural Model for health service utiliza-tion [13] to explore the factors that influenced expectant mothers' decisions to deliver at home. According to Andersen and Newman [13] an individual's use of health services is a function of three important interrelated factors—predisposing, enabling/inhibiting and need factors [13]. The predisposing factors include socio-cultural characteristics of individuals such as education, occupation, ethnicity, culture, health beliefs, values, health status, and knowledge that people have about the health system, and demographic factors such as sex and age [13]. The enabling/inhibiting factors refer to the resources available or otherwise, that enable or inhibit the individual's ability to use health services. The fact is that individuals may be predis-posed to use health services but may lack the means or resources to enable them do so [14]. Enabling factors are proxied by individual or family resources such as income, health insur-ance coverage, and community level factors such as, the availability of health personnel and facilities. The need factors represent the most immediate cause of health service use [14]. Two important types of need factors arise here—perceived need and evaluated need. Perceived need refers to, "how people view their own general health and functional state, as well as how they experience symptoms of illness, pain, and worries about their health and whether or not, they judge their problems to be of sufficient importance and magnitude to seek professional help" [14]. Evaluated need represents professional judgment about people's health status and their need for medical care [14]. Our study therefore posits that the interaction of these factors —predisposing, enabling and need factors, within the peculiar institutional and socio-cultural contexts of childbirth, influence women's decisions to deliver at home.

## Materials and methods

### Study setting

The study was conducted in the Jirapa Municipality of the Upper West Region of Ghana, between November 2019 and June 2020. According to the 2010 Population and Housing Cen-sus of Ghana, the Jirapa Municipality has a population of 88,402, representing 12.6 percent of the Region's total population. Females constitute 53% of the population, yet only 48% of the females are literate [7]. The Municipality has 44 health facilities, comprising seven health

centres, 35 functional CHPS, one Polyclinic and a hospital—the St. Joseph's Hospital. The St. Joseph's Hospital is managed by the Catholic Church and serves as a referral point for the health centres and CHPS in several communities within and outside the Municipality [15]. The hospital has six medical doctors and 43 midwives [9]. Between 2014 and 2017, the Jirapa Municipal Hospital (the St. Joseph's Hospital) recorded fifteen maternal deaths—five in 2014, three in 2015, two in 2016 and five in 2017 [9, 15]. Data available showed that in 2017, about 2,359 expectant mothers registered for ANC and 58.1% of them attended at least one ANC visit. In 2018, 2,398 expectant mothers registered for ANC, of which 58.3% attended one ANC visit [15]. Of this figure, 1,600 women registered for ANC in the first trimester [15]. Fig 1 details a map of Ghana showing the Jirapa Municipality and the communities, where the study was conducted.

## Study design, sampling and recruitment

The study adopted a qualitative, exploratory, cross-sectional research design. It targeted pregnant women who received antenatal care during pregnancy but delivered at home in the Jirapa Municipality. The study also included key informants such as traditional birth attendants (TBAs), midwives, husbands, opinion leaders such as assembly members, (assembly members are elected representatives of their electoral areas or communities at

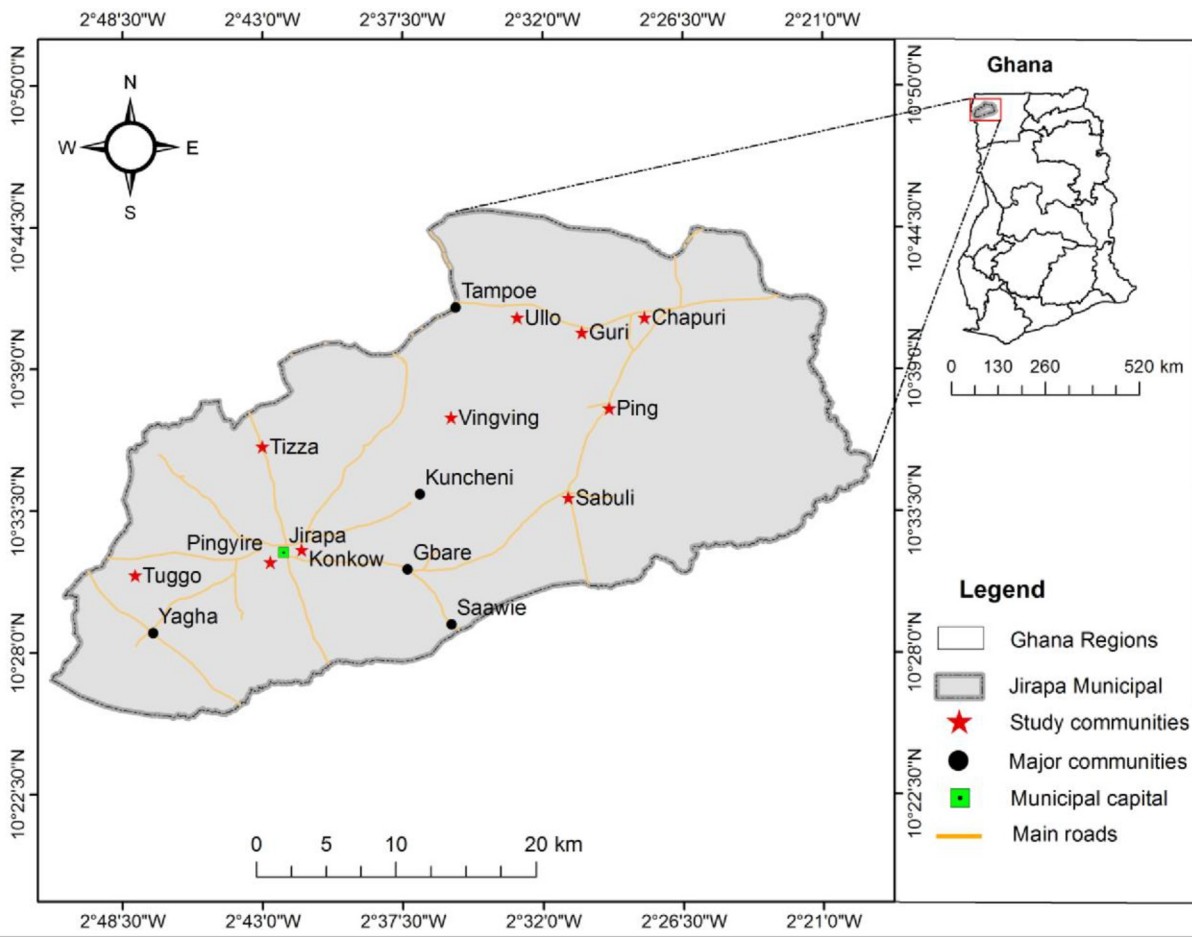

**Fig 1. Map of the study area.**

the District Assembly level) and chiefs. We obtained the list, and telephone numbers of 115 women who attended ANC at the St. Joseph Hospital but delivered at home. The list was extracted in July 2019. We purposively selected 23 women from this list, who were identified to have delivered their babies at home. The ages of the women ranged between 18–40 years. Eighteen of the women (78%) did not have any formal education and five of them (22%) had primary education. Purposive sampling was also used to select seven husbands, 10 midwives at the St. Joseph Hospital, three TBAs, three chiefs and four Assembly members, bringing the total sample size to 50 respondents. Two of the men were husbands of the sampled women who delivered at home, while the five were other men. We purposively wanted to appreciate the views of the husbands of the sampled women and other men for the purposes of data triangulation. Based on a preliminary review of the data from these 50 respondents, by reading the field notes and listening to the tapes, we realised that no additional information was added from further responses —thus data saturation was reached with the 50 respondents [15]. All the participants in the study were native Dagaabas. All sampled respondents were contacted through phone calls and only those who consented to participate in the study on phone were recruited into the study.

## Data collection and analysis

Semi-structured interviews were conducted with the 50 research participants, using an interview guide. The interviews were conducted at the homes of the respondents in a serene environment, free from distractions. The interview guide covered respondents' views on how socio-cultural and institutional level factors influence women's decisions to deliver at home. Four female graduate research assistants were recruited and trained to collect the data. As a quality control measure, the interview guide was pre-tested. Member-checking was also used to further enhance the quality of the data—where the data collectors read the recorded responses to the respondents for validation purposes. The interviews with the midwives and assembly members were conducted in English, while those with the other respondents were conducted in the local dialect (Dagaare). The interviews lasted averagely between 40 minutes and 1 hour. All the interviews were audio recorded and transcribed. We validated the English for interviews conducted in the local dialect by back translating the interview guide in the local dialect in English by a language expert—a bilingual in both Dagaare and English. The transcripts were checked by the authors to ensure consistency in transcription and translation into English.

The data were manually coded and analysed by the first and second authors and validated by the third author. A five- staged analytical procedure was deployed in analysing the data— namely; familiarisation, identification of themes, indexing, charting, and mapping and interpretation [16, 17]. At the familiarisation stage, the second author listened to the audio tapes and thoroughly read through the transcripts, to ensure that the transcripts were consistent with the audio tapes. In the second stage, the first and second authors individually coded the transcripts, and discussed the emerging themes with the last author, who read only half of the transcripts. During indexing, the first and third authors categorised the themes in consonance with the analytical procedure of the study. In the stage of charting, the first and second authors summarised the data according to the themes. In the final stage of mapping and interpretation, all three authors reviewed the themes and made connections between them to bring out similarities and differences, in order to interpret the data. This procedure facilitated the development of themes both deductively and inductively from the narratives of research participants [15, 16].

### Ethics statement

The Navrongo Health Research Centre (NHRC) Institutional Review Board (IRB) granted ethical approval (NHRCIRB401) for the study. The Upper West Regional Health Directorate and the St. Joseph Hospital in Jirapa, granted us permission to conduct the study. Written, informed consent was obtained from each respondent before the administration of the data collection instrument and audio-recording of the responses.

## Results

### Predisposing factors of home delivery

**Traditional practices—home birth, a sign of faithfulness to husbands.**   Our study revealed that traditional practices such as the need for a woman to prove her faithfulness to her husband predisposed women to deliver at home. Delivering at home, socio-culturally is seen as a symbol of faithfulness to husbands in the study context. The cultural dictates of the studied population postulate that a woman who is faithful to her husband should be able to deliver naturally (i.e. spontaneous vaginal delivery), without any form of assistance from the health facility. For this reason, any woman who is given any form of injection, forceps or vacuum to help bring out a baby is considered unfaithful—which is why she could not have a spontaneous vaginal delivery. As a result, the majority of women would want to endure the struggle and deliver at home, to prove their faithfulness to their husbands rather than going to deliver at the hospital. In addition, it is a revered traditional belief that, as a sign of marital fidelity, no man is allowed to see the private part of a married woman except the husband, to the extent that women could lose their marriages when for example, a male health professional sees their nakedness. Delivering at home, therefore, prevents women from being exposed to other males.

> *"Some husbands always argue that women are unfaithful to their husbands if they deliver at the hospital. A woman came to deliver in this facility but due to her baby's breach presentation and weight, she could not have normal delivery, which compelled the staff to arrange her for caesarean section and when the husband was called to the facility and asked to thumb print to enable the doctors carry out the procedure, the man refused to thumb print and said he never knew he was living with an unfaithful woman"* (Midwife, St. Joseph's Hospital, Jirapa).

> *"I am compelled to deliver at home, because, in this village, husbands always see you as someone who has ever committed adultery in the course of your marriage, if you deliver in the hospital"* (Woman, 32 years).

> *"According to my husband's tradition, it is not proper for another man to see the private part of a married woman, since that will mean the woman is spoilt and hence, he can't marry her again"* (woman, 18 years).

*Religious beliefs.* Also, some religious beliefs dictate that the 'gods forbid' hospital delivery, thus compelling some women to deliver at home. Some respondents narrated that the gods cannot be disobeyed and that those women who disobeyed the gods, to deliver at health facilities will have negative birth outcomes—stillbirths for example because the gods are claimed to have supreme power, relative to the medicines provided at health facilities. In fact, even women who preferred to deliver at health facilities are restrained by their mothers in-law from doing so, on the basis of the supremacy of their gods. The narratives below attest to this assertion:

*"It is sometimes very scary to say that you will not obey what your husband tells you. A woman once disobeyed the husband who said their house gods forbid hospital delivery and had her first two children delivered at the hospital, unfortunately both ended in stillbirths. Thereafter, she decided to have the subsequent deliveries at home which all resulted in live births. Having witnessed this scenario, I'm not prepared to allow this happen to me, so all my deliveries will be at home"* (Woman, 40 years).

*"I had wanted to deliver in the hospital but I was told by my mother in-law that all her eleven (11) children, including my husband were all born in the house, because the gods they worship are stronger than the white man's medicine (modern healthcare)"* (woman, 23 years).

*Myths about consequences of hospital delivery*. Our results also revealed myths surrounding hospital delivery including the perception that children born in the hospital are weaklings, and some of them also die mysteriously. These perceptions, which are passed on from one generation to another, influence the majority of women to deliver at home, in order not to lose their babies, even during complicated labour.

*"In this community, children delivered in hospitals are considered to be weaklings and are not as strong and healthy, as compared to those delivered at home by our grandmothers. Because of this, a majority of us have refused to go to the hospital to deliver"* (Woman, 25 years).

*"Madam, all members of this family give birth in this house no matter how complicated the labour is. This is because from our elders, the first three women from this family who delivered in the hospital lost their babies under very strange circumstances, whilst from time immemorial, those who are delivered in the hands of someone inside this house do not encounter any problem"* (woman, 28 years).

*Women's autonomy in the health decision-making process*. Our results illustrate that women's autonomy in household health- seeking decision-making processes is a key predisposing factor of home delivery. Women's autonomy is linked to their socio-economic status. Many women in the rural areas are poor, and may not be able to oppose their husbands' decisions as to where they have to deliver. As the narratives below explain, some of the women cannot even challenge their husbands' decision to deliver at home, despite the difficulties they experience when in labour because they are not economically empowered—they do not have the financial resources to challenge their husbands' authority and deliver at the hospital. Because of the vulnerable nature of the women, they respect their husbands' views, in order to stay peacefully in their marriages. Three women narrated their experiences regarding their autonomy in health decision-making as follows:

*"Please this issue should be discussed with my husband because he will know better. I am saying this because, the last time I argued with him about the reasons why he always compels me to go through all this pain to deliver in the house instead of our district hospital, he angrily retorted that since I have come of age and now ready to question his authority in the house, I should pack my things and go to my fathers' house* (Woman, 22 years).

*"Sister, I don't want any trouble with my husband who is the head of the family, he is the one who gives me money, food and also provides shelter. As you can see, I don't do any kind of income generating work. All I do is to fetch water, cook, sweep and do other basic things in the house, so if he insists that I should give birth in the house, I will do it for the sake of my marriage and children* (Woman, 30 years).

*Illiteracy*. Our research found that illiteracy predisposes most women to deliver at home because they usually forget the expected delivery dates recorded in the ANC books, as they are unable to read the information. As the Assemblyman narratives below, it is not only some of the wives who are illiterates but also their husbands. Because of the high levels of illiteracy, the husbands especially, do not value the importance of hospital delivery.

*"We resort to delivering at home because we cannot read what is always written in the antenatal books; so, we end up forgetting the expected date of delivery, and are unable to prepare to go to the hospital before labour starts"* (Woman, 27 years).

*"Madam, it is not only the wives who are not educated but their husbands as well. The majority of husbands cannot read and write; they don't know the importance of delivering in the hospital. How do you expect such a person to always encourage the wife to go to the hospital for delivery?"* (Assemblyman, 45 years).

## Enablers of home delivery

**Rude behaviour, poor treatment and negligence by healthcare providers.** Our study established that rude behaviour, poor treatment meted out to pregnant women and negligence from healthcare providers during ANC and delivery visits deter expectant mothers from delivering at the health facility and therefore enable home delivery. This unfriendly attitude of midwives makes women prefer to deliver at home, under the supervision of TBAs. Women fear they will lose their lives when they go to deliver at the health facility due to the rude and unconcerned attitudes of some midwives. Two women remarked as follows:

*"Please madam, it is not that we prefer the traditional birth attendants, oh no!!! but the negative feedback we get from our colleagues about the poor treatment meted out to them in the hospital compels us to do so. From our friends who have delivered in the hospital before, health workers keep on insulting them for the small mistake they make". (*Woman, 28 years).

*Madam, the truth is that I used to attend ANC with a neighbour whose pregnancy was more advanced than mine. When she went into labour and she was taken to the hospital, the nurses were so rude and unconcerned: the health workers were there chatting and playing with their mobile phones, instead of attending to her. So, she passed on; so, tell me, if you were me, will you go to that same hospital to deliver*? (woman, 34 years).

**Inadequate health professionals at health facilities.** Some respondents alluded to the inadequate number of midwives in the various health facilities as a disincentive for most women who intended to deliver in health facilities. They reported that even though there are midwives and other cadres of health workers currently at the facilities, their numbers were woefully inadequate to attend to the overwhelming number of women in labour at any particular point in time. Because of the inadequacy of midwives and doctors, some women do no appreciate the value of health facility delivery, when consideration is given to inconveniences such as the waiting time. The quotes below buttress this point:

*"Sometimes it's not that they don't come here to seek delivery services in the facility. The number of health workers who are here to attend to them is not adequate. For instance, last week I was alone on duty when four women were rushed in for delivery. By the time I was done with the first one, a second one had delivered on the floor. Do you expect such an unfortunate woman to ever come to the hospital again to deliver*? (Midwife, St. Joseph Hospital).

*"My firstborn died in the hospital because the medical doctor who should have come to per-form the caesarean section delayed. So, I said to myself, 'what is the value of going to deliver in the hospital with all its attendant inconveniences when the outcome will not be good?' Because of this, I decided to give birth in the house, since my mother is also a traditional birth atten-dant"* (Woman, 39 years).

**Lack of privacy and confidentiality at health facilities.** Our study found that the lack of privacy and confidentiality in hospital settings compelled many women to deliver at home. The issues of privacy and confidentiality serve as roadblocks for male involvement in care delivery because there are no private rooms for women to deliver, thus, making it uncomfort-able for men who want to be present when their wives are in labour. Some of the respondents even alleged that some health workers disclosed the HIV/AIDS statuses of pregnant women who tested positive during ANC visits to their friends. The narratives below detail these claims:

*"My sister, despite the fact that I am a man, I realize women suffer during labour. As such, I always want to be closer to my wife when she is in labour. However, at the hospital, you have about five or six women in the same room labouring, how can you be comfortable under such circumstances? I simply prefer the house and for that matter my room, where I can easily have access to my wife any time I so desire"* (Husband, 45 years).

*"In the last month, my friend who is a midwife showed me a pregnant, 18-year-old girl, who was tested positive during their STIs screening. As a result of this, I have decided not to go there for my next ANC, not to even talk about delivering in the hospital"* (Woman, 21 years).

**Hidden charges for out-of-pocket payments during delivery.** Interestingly, a majority of the women interviewed reported that they are sometimes charged illegal fees contrary to the free maternal health care policy. Respondents argued that midwives asked them to buy too many delivery kits, some of which the midwives appropriate to themselves. These extra out-of-pocket charges embarrass women who cannot afford, and run counter to the free maternal health care policy. Because the services of TBAs are free or do not involve significant costs, some women prefer to deliver under the supervision of TBAs, relative to delivering at the health facility. Two respondents during the interviews noted that:

*"Please, it's very annoying delivering in the hospital. They always say that it's free but that's not true. The last time I delivered in the hospital, they [Midwives] kept demanding that I buy Dettol, soap, and other detergents, which I didn't have the money to buy and so I felt embar-rassed".* (Woman, 38 years).

*In fact, there is nothing in the delivery room, so you need to buy everything including the delivery bed. I don't have money to buy all the things the hospital will need, but with virtually no money or a little as just five Ghana Cedis (GHS 5), is enough for the traditional birth atten-dants and I will have my baby* (Woman, 40 years).

**Distance to health facilities.** Access to health facilities from place of residence remarkably determines where delivery takes place, as contained in the narratives below. Some of the women interviewed actually wished to deliver at the hospital but were unable to do so due to the bad road network that makes transportation a big challenge. There are no ambulances readily available to transport pregnant women in labour to the hospital. The deplorable nature of the roads in the study area further makes it difficult for private vehicles to ply the roads

frequently, and the associated cost of transportation is a disincentive for some women in the rural areas who want to deliver in the hospital. The quotes below support this point:

> *"I would have wished to deliver in the hospital but all my attempts have always ended on the way to the hospital. This is because each time my labour starts, the only community vehicle is not available and the district ambulance is also broken down. By the time help eventually arrives, it is often too late" (*Woman, 26 years).

> *"As for me, I would have preferred that my wife delivers at the hospital but we have just one "Nyaaba Lorry" (tricycle), which often goes to town and returns late. Since this is our only source of transportation*, *if you need it at the time when it is away what do you do*? (Husband, 35 years).

## Need-based factors influencing home delivery

**Fear of caesarean section or assisted delivery.** Fear of caesarean birth or assisted delivery creates the need for home delivery. Stories about health workers leaving needles or other material used during caesarean sections in the wombs of women, scare some of the women from going to deliver at the hospital. Some husbands also indicated that caesarean birth weakens women, such that these women may not be able to effectively perform their household chores or help their husbands on the farm. From their experiences, some husbands also see caesarean sections as risky undertakings—they are a matter of life and death, and would not risk the lives of their wives to undergo an operation. The three quotes below vividly illustrate this assertion:

> *"Please it's very sad to go to the hospital and end up being operated upon. From what I have heard from neighbours, sometimes the health workers may forget a needle or materials in your womb, so I prefer home delivery to hospital delivery, no matter how complicated it is" (*Woman, 24 years).

> *"For me*, *I generally will let my wife deliver at home*, *where the option of operation will not arise at all. She does all the household chores and some sometimes helps me on the farm. Imagine she is operated upon; how can I manage all these things by myself. The operation not only keeps her away from work for a long time, but it also weakens her"* (Husband, 35 years.)

> *"My sister, we are guided by our past. My friend's wife walked into the theatre for the operation but her lifeless body was brought back. As a result, I prefer home delivery because I am not ready to risk my wife's life. Life is only one"* (Husband, 39 years).

## Discussion

Drawing on the Andersen and Newman's Behavioural Model, this paper explored the important question: Why do women attend ANC but deliver at home? Our study revealed a number of intriguing factors—socio-cultural and institutional level factors that explained why some women attended ANC but ended up delivering at home. Our analysis established that socio-cultural factors—faithfulness to husbands, religion, and illiteracy predisposed most women to deliver at home. Our results demonstrate the community's strong affinity to their traditional belief system, believing in the power of their gods to guarantee safe pregnancy and delivery. Our results are consistent with earlier studies on the subject matter in Ghana and elsewhere. For example, Barbi *et al's*. [18] study in the Volta region of Ghana reported that socio-cultural practices and beliefs were obstacles preventing women from ANC visits because they believed

that God will take care of them throughout the period of pregnancy and delivery. In Nepal, Paudel *et al's*. [19] paper reported that the health- seeking behaviour of families is a function of their belief in God's will in disease and death and that God has the capacity to affect their lives both negatively and positively. Similarly, studies in Sierra Leone, Liberia and Zimbabwe reported that obstructed labour was considered a sign of infidelity to one's husband [20]. To prove their fidelity to their husbands, or bring honour to both husbands and families, many women prefer to deliver at home without professional assistance. These strongly- held traditional beliefs have implications for the use of maternal health services in general, and delivery services in particular. Indeed, other studies in the Tallensi District in the Upper East region of Ghana, attributed women's non-use of maternal health care services to strong traditional belief systems [21]. These traditional beliefs could be attributed to the low educational status of women and their husbands in the study context. Studies in Pakistan, Ethiopia and Guinea-Bissau reported that educational status was a major predisposing factor associated with health facility delivery. For instance, existing quantitative studies [22–24], reported that women without formal education were more likely to deliver at home compared to women with higher levels of formal education. These findings demonstrate the empowering effect of education on women. Women with higher levels of education are more likely to have increased knowledge of the benefits of health facility delivery, increased socialisation to interact with formal services outside the home environment, familiarity with modern medical culture, and access to increased financial resources. At the same time, husbands with higher levels of education are more likely to facilitate their wives' motivation to deliver at the health facility [25].

Women's autonomy also stood out prominently as a predisposing factor associated with home delivery, largely based on their weak economic position. The women interviewed depended entirely on their husbands' largesse, and as such would not challenge their husbands' position as to where to deliver, in order to secure their marriages. These findings fit in neatly with Ameyaw *et al's*. [26] study in Ghana, that reported that women with health decision-making autonomy have higher chances of health facility delivery, as compared to those who are not autonomous. Lowe *et al*. [27] and Kifle *et al*. [28], in their research in Gambia and Eritrea respectively also found that women's decision to receive care by trained personnel during delivery was beyond their control because women do not have control over material and financial resources in the household; and that even women with higher educational levels may fail to translate their preference for a delivery place into actual behaviour if their husbands are opposed to their choice. Kifle *et al*. [28] argued that a woman's choice to deliver at a facility is seriously undermined by women's lack of decision-making autonomy through complex processes of gender inequality. Kifle *et al*. [28] assert that family decision-making power regarding the use of maternal healthcare services is strongly influenced by the values and opinions of husbands, mothers-in-law and other close relatives.

Remarkably, this study established that institutional-level factors—rude behaviour, negligence, general poor attitude of health professionals, inadequate midwives, lack of privacy and confidentiality, and hidden charges at health facilities, played a double role of enabling some women to deliver at home and at the same time acted as obstacles to health facility delivery for others. Some health providers are reportedly rude, negligent, and unfriendly towards women during ANC and delivery visits [29, 30]. In fact, the issue of negligence is particularly worrying because women in labour are left unattended, while midwives played with their mobile phones. These issues relate directly to the care environment, where women give birth, regarding the perceived quality of care received. The perceived quality of care is an important determinant of health services use. Perceived quality of care increased the number of women giving birth in health facilities, and women even travelled farther than expected to give birth at facilities, where the quality of care is perceived to be good [31]. The perceived quality of care further

speaks to the issue of 'respectful maternity care (RMC)'—defined as friendly and woman-centred [32]. Hajizadeh *et al.* argued that RMC is a fundamental human right that includes respecting women's beliefs, independence, emotions, dignity and preferences [32]. The authors observed that there is a direct relationship between RMC and positive childbirth experience, and that disrespect and abuse violate the basic principles of ethics, human rights, and basic obligations in providing care for patients [32]. Like the women's account in this study, Bulto *et al's.* [33] study in Ethiopia revealed that the proportion of women who received RMC during labour and childbirth was low (35.8%) and recommended monitoring and reinforcing accountability mechanisms for health workers to improve RMC [33].

The inadequate number of midwives reported in this study is perhaps, attributable to the considerable distributional inequalities in clinical health staff within and across all the regions in Ghana. For example, in 2018, out of 5,582 midwives produced, the Upper West Region, which is more rural in outlook, got only 219 representing 3.9%, whereas the more urbanised regions of Greater Accra and Ashanti had 973 and 1,281 representing, 17.4% and 23% of midwives respectively [34]. The irony of this situation, is that the Jirapa Municipality, where this study was conducted, has a reputed midwifery training school. Consistent with the existing evidence, studies in Nepal reported that poor health infrastructure, and inadequate nurses/midwives were critical factors associated with home delivery [35]. The issue of lack of privacy and confidentiality, to the extent that unauthorised personal information of women, relating to their HIV/AIDS statuses is given to third parties, is an indictment on health staff, and a violation of the ethical code of conduct because the divulged information could lead to discrimination and stigmatisation against the affected women. The lack of privacy in terms of adequate accommodation and space in the labour room for husbands and relative to have access to a woman in labour, has also been reported in earlier studies in Ghana and elsewhere as a barrier to health facility delivery [34, 36]. More intriguing is the revelation that it is not free to deliver at the hospital, as women in labour incur several hidden costs, including buying Dettol, soap and other detergents because the delivery room according to the narratives, has "nothing". The belief that some of the midwives appropriate some items bought by the women in labour, speaks to corruption in the health sector. These hidden costs run contrary to the FMC policy under Ghana's flagship NHIS and question the effectiveness of the policy because the policy clearly enunciates free access to maternal healthcare services. Studies in the Kassena-Nankana Municipality earlier reported that, the implementation of the NHIS did not eliminate all charges and that women still made out-of-pocket payments for ANC services [37–39]. Financial constraints have been cited as one of the major factors for home delivery, while the reverse holds true—that is, women from wealthy homes have higher odds of seeking health facility delivery [18, 40]. Indeed, it is argued that the rise in maternal deaths, especially in developing countries, may be associated with increasing costs that act to delay the decision to use the hospital until the woman's condition is critical [19].

Lack of, and cost of transportation to the nearest facility promoted home delivery. Most of the respondents relied on tricycles as the only means of transport. Thus, the farther away respondents reside from the health facility, the more likely they are to deliver at home rather than delivering in a hospital. Most of these women are reluctant to deliver in the hospital because they do not readily have means of transport when labour starts. Our findings resonate well with earlier studies. For example, Dotse-Gborgbortsi *et al's.* [41], study in the Eastern Region of Ghana, reported that a kilometre increase in distance significantly reduced the prevalence of women giving birth in health facilities by 6.7%. The distance is compounded by the timing of labour, especially in the night. A study in the Upper East Region of Ghana reported that when labour starts at night, women could not walk to the health facility to deliver [21].

Finally, the fear of caesarean section or assisted delivery has been established in this study as the only need-based factor influencing home delivery. The fear of caesarean section or assisted delivery may be appropriately placed under institutional level factors because it occurs only at the health facility or institution. As the results suggest, it appears most of the respondents have a fear for assisted delivery because health professionals may become negligent and forget needles and other materials in the wombs of women, the perception that the procedure may weaken women for life, and death. Earlier scholarship from rural Bangladesh also reported that when expectant mothers go to health facilities, the doctors hurriedly conducted caesarean delivery instead of trying for a normal vaginal delivery. Based on these fears, some women prefer to deliver at home [42].

## Conclusion and recommendations

This paper investigated the factors that influenced women to deliver at home, despite receiving ANC at a healthcare facility. The paper established that socio-cultural and institutional level factors influenced women's decision to deliver at home. Socio-cultural factors such as faithfulness to husbands, religious beliefs, women's autonomy, and the lack of, and cost of transportation, predisposed most women to home delivery. Institutional- level factors such as rude, negligent and unfriendly behaviour from health professionals, inadequate midwives, lack of privacy and costs, and the fear of caesarean sections, were found to be crucial barriers for health facility delivery. Our findings have far- reaching policy implications for increasing health facility delivery and reducing maternal and child mortalities, in consonance with SDG Three. We recommend that, given the low levels of literacy and the strong traditional beliefs in the study area, the Ghana Health Service should collaborate with civil society organizations and community leaderships to organize educational programmes, aimed at sensitising communities on the importance of health facility delivery, and reshaping some of the traditional belief systems facilitating home delivery. Additionally, women empowerment should be promoted, to enhance their decision-making autonomy relating to health seeking. We also recommend that Ghana Health Service recruits and posts more midwives to the study area and give them incentives, to motivate them to stay in these deprived areas. Finally, the Ghana Health Service should also equip the labour rooms in health facilities, with the requisite delivery kits to facilitate the work of midwives, to give meaning to the FMHP.

## Author Contributions

**Conceptualization:** Kennedy A. Alatinga, Jennifer Affah, Gilbert Abotisem Abiiro.

**Data curation:** Jennifer Affah.

**Formal analysis:** Kennedy A. Alatinga, Jennifer Affah, Gilbert Abotisem Abiiro.

**Methodology:** Kennedy A. Alatinga, Jennifer Affah, Gilbert Abotisem Abiiro.

**Writing – original draft:** Kennedy A. Alatinga.

**Writing – review & editing:** Kennedy A. Alatinga, Jennifer Affah, Gilbert Abotisem Abiiro.

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
