## [Decision Letter · Decision Letter 0]

15 Jul 2021

PONE-D-21-15389

Exploring factors influencing home delivery: stakeholders’ perspectives from a rural Ghanaian District

PLOS ONE

Dear Dr. Alatinga, 

Thank you for submitting your manuscript to PLOS ONE. After careful consideration, we feel that it has merit but does not fully meet PLOS ONE’s publication criteria as it currently stands. Therefore, we invite you to submit a revised version of the manuscript that addresses the points raised during the review process.

We look forward to receiving your revised manuscript.

Kind regards,

Gouranga Lal Dasvarma, PhD

Academic Editor

PLOS ONE

Journal Requirements:

2. Please include a copy of the interview guide used in the study, in both the original language and English, as Supporting Information, or include a citation if it has been published previously.

5. Please ensure that you include your title page within your main document.

Additional Editor Comments (if provided):

Dear Dr. Alatinga,

Thank you for submitting your manuscript for consideration for publication in PLOS One.

Please revise the manuscript as required by the reviewers. Please have the manuscript edited for English even though the reviewers have not indicated so, and submit the revised and edited manuscript within 30 days of receiving this letter.

Reviewers' comments:

Reviewer's Responses to Questions

**Comments to the Author**

1. Is the manuscript technically sound, and do the data support the conclusions?

Reviewer #1: Yes

Reviewer #2: Yes

2. Has the statistical analysis been performed appropriately and rigorously? 

Reviewer #1: N/A

Reviewer #2: N/A

3. Have the authors made all data underlying the findings in their manuscript fully available?

Reviewer #1: Yes

Reviewer #2: Yes

4. Is the manuscript presented in an intelligible fashion and written in standard English?

Reviewer #1: Yes

Reviewer #2: Yes

5. Review Comments to the Author

Reviewer #1: Thank you for providing this opportunity to comment on this paper. I felt great to be informed of the contexts and could very well relate the findings from this paper with my recent study in the related field conducted in mountains of Nepal. It very well resonates and adds to the argument that childbirth needs to be viewed from socio-cultural lens. It also equally voices the need to strengthen health system accountability for service readiness. Only then the quality of care and safety (health as well as cultural safety) during childbirth can be met, and available services can be utilised.

Introduction

The section is well written and provides clear gap of understanding how the interactions of institutional and socio-cultural factors impact in women’s decision making. Authors have also clearly presented need for the study and their position ‘Andersen and Newman Behavioural theoretical model for health service utilization’. This is great, which we do not commonly find in published papers.

Methods

Study setting

• Authors can visualise study setting in country map. This might well locate international readers while reading the paper.

• Authors can shorten this section by only bringing key stats around HDI, female literacy and local health system context (which they have done in second paragraph). You might want to bring local targets around expected pregnancies and births—the proportion of women expected to give birth each year, and stats on at least 1 ANC visit.

Study Design

• Authors can just keep the first sentence about the type of the design, the rest in the paragraph can be omitted as it basically adds no more new information.

• Study Population and Sampling: I have asked couple of questions below just to add in the quality of the methodological process implemented:

• Authors might want to reword it ‘study population, sampling and recruitment’.

• Can authors clarify how many women were in the list received from the St. Joseph? And, which year or months the list was extracted and why?

• What were the criteria to recruit the 7 husbands purposively? Were these the husbands of the women interviewed? Or other husbands?

• How many total participants did the authors approach? Were there any refusals?

• Can authors present a table with key socio-demographic characteristics (age, sex, ethnicity, number of ANCs in the last pregnancy, number of pregnancy, no of babies, place of previous childbirth, years of experience for midwives etc) of the participants from each group? This will provide a much better sense to the readers.

• Did authors provide any incentives to these participants?

• How did authors figure out that data saturation was reached at 50 respondents? Adding a few sentences about your practical insights might help future researcher, as this is mostly not a straightforward decision in the field. When did authors realise this—while collecting data in the field, at analysis stage?

• Write one or two sentences on how you validated English for the interviews taken in the local dialect.

• Can authors provide the semi-structured interviews as additional file?

• Can authors also provide the thematic framework and the initial coding structure they used as additional files?

• While 4 research assistants were used in data collection, how many were involved in coding and analysis?

• As authors said the framework used helped them to develop themes deductively—Can authors discuss this to what extent they were led by it? Did authors come up with any participants data which can challenge in any way to the already established framework they deployed?

• If the matrix authors created is not too bulky, they can include that as an additional file

Results

Overall, the quotes presented provide quite powerful picture of why women end up giving birth at home. Several quotes author presented show adequate thematic prevalence. It is so interesting to read that all women indeed want to come to hospital. But, the lack of trust in facility births (especially operations and assisted births), faithfulness towards husbands, poor quality of facility birth (privacy, inadequate staff) are key factors which sustain home births.

Some specific comments below:

• Authors said “For this reason, any woman who is given any form of injection, forceps or vacuum to help bring out a baby is considered unfaithful—which is why she could not deliver naturally on her own”---did authors mean she could not decide to seek healthcare on her own?

• In the first quotation, ‘breath presentation’—did the health worker mean ‘breach presentation’?

• Can authors write ‘Midwife and years of experience’ in the parenthesis if the health worker they meant is a midwife? You might not need to write ‘St. Joseph’s Jirapa’ as every health worker you interviewed are from here?

• Likewise, authors can simply write ‘Woman and her age’, no need to repeat delivered at home if authors have interviewed only those who delivered at home. If available, authors might want to specify primi or multi.

• What did husbands say on the ‘sign of faithfulness’? any quotes from husband??

• The quotation on the religious belief you provided does not fully establish gods forbid, it rather establishes that women have indeed gone to hospital but did not have intended outcomes. What was the reasons behind her fresh stillbirths—delay in receiving care? Poor quality of intrapartum care? How common is this theme in your study? It would be great to support this with some additional quotes.

• Do people live in joint family or nuclear family in the study setting? How common is husbands’ accompaniment to their wives during ANC check-up?

Discussion

It is a great discussion. Yet, I would recommend authors to shorten it and discuss only those unique findings---interaction of religious beliefs and hospital births as sings of infidelity to their husbands and gods; women’s autonomy and the complex family context; lack of trust and poor satisfaction with health facility births; and the policy factors such as free childbirth policy. Authors might want to check my papers if it supports them in discussion in anyway: https://journals.plos.org/plosone/article?id=10.1371/journal.pone.0194328;
https://bmcpregnancychildbirth.biomedcentral.com/articles/10.1186/s12884-018-1776-3; and https://jhpn.biomedcentral.com/articles/10.1186/s41043-018-0148-y

While discussing the poor satisfaction and trust with the quality of care in facility births, authors might want to refer to literatures related to ‘respectful maternity care’.

To save some words, authors might want to omit this from their first paragraph in the discussion section—“These strongly held traditional beliefs could perhaps, be attributed to the low educational status of women and their husbands in the study context. Because of the low educational levels of women and husbands, the women are not capacitated to detect when labour sets in while their illiterate husbands do not even appreciate the importance of delivering in the hospital. These findings corroborate earlier studies in Pakinstan, Ethiopia and Guinea-Bissau, which reported that educational status was a major predisposing factor associated with health facility delivery. For instance, existing quantitative studies, (26– 29) reported that women without formal education were more likely to deliver at home compared to women with higher levels of formal education. These findings demonstrate the empowering effect of education on women because women with higher levels of education are more likely to have increased knowledge of the benefits of health facility delivery, increased socialisation to interact with formal services outside the home environment, familiarity with modern medical culture, and access to increased financial resources. At the same time, husbands with higher levels of education are more likely to facilitate their wives’ motivation to deliver at the health facility (30).”

Conclusion and Recommendations

Authors can save words by omitting “The paper found very interesting and policy relevant results, including faithfulness to husbands, religious beliefs, women’s autonomy, and lack of, and cost of transportation as the major factors that predisposed most women to deliver at home. Institutional levels factors such as rude, negligent and unfriendly behaviour from health professionals, inadequate midwives, lack of privacy and confidentiality and hidden costs, and fear of caesarean delivery were found to be crucial barriers for health facility delivery. Our findings have far reaching policy implications for increasing health facility delivery and reducing maternal and child mortalities in consonance with SDG three.” From the conclusion and recommendation section.

Thank you for the great work.

Reviewer #2: The authors have done a qualitative study to understand some socio-cultural and institutional factors that inhibit women from giving birth to child in hospital. This is a well argued article and can be accepted with a revisions on the following :

1. The authors organised the findings part in three major sections according to the given theoretical framework. However, in some subsections of the findings not adequate explanation of data, i.e., no or least explanation of the quotes is given. Only presenting quotes do not suffice to make the data presentation up to the standard. I suggest authors to revise the whole findings part, draw the findings and explanations from the quotes presented.

2. A brief discussion of socio-demographic information of the mothers interviewed need to be added in the methodology section. It can be added in the sub section 'study population and sample'

3. The format of the quote, especially who gives the quote, needs to be uniformed. for example for women's quote it is written (Woman who delivered at home) and for husbands' quote (IDI with husband). Patterns need to be uniformed.

6. PLOS authors have the option to publish the peer review history of their article (what does this mean?). If published, this will include your full peer review and any attached files.

Reviewer #1: **Yes: **Dr. Mohan Paudel

Reviewer #2: **Yes: **Sanzida Akhter

---

## [Author Response · Author response to Decision Letter 0]

29 Sep 2021

Response to reviewers

We thank the editor and reviewers for evaluating our manuscript. We provide point by point responses to the comments raised by the editorial team and the reviewers. Please, find our responses to every comment/question in the table below. We have highlighted the revisions in tracked changes in the main manuscript

PONE-D-21-15389:

Comments Authors’ Response

Editorial Comments

 We thank the editorial team for graciously drawing our attention to these important requirements. We have revised the manuscript in line with the journal’s style requirements.

2. Please include a copy of the interview guide used in the study, in both the original language and English, as Supporting Information, or include a citation if it has been published previously.

 We thank the editorial team for this request. We have now included the interview guide in both the original language and English, as Supporting Information.

3. We note that you have indicated that data from this study are available upon request.

b) If there are no restrictions, please upload the minimal anonymized data set necessary to replicate your study findings as either Supporting Information files or to a stable, public repository and provide us with the relevant URLs, DOIs, or accession numbers. For a list of acceptable repositories, please see http://journals.plos.org/plosone/s/data-availability#loc-recommended-repositorie

 Because this is purely a qualitative study, the data was in the form of hand written notes and the transcripts. All the data in the form of the transcripts are also used as the direct quotations in the manuscript so there is no data left to be uploaded.

4. PLOS requires an ORCID iD for the corresponding author in Editorial Manager on papers submitted after December 6th, 2016. Please ensure that you have an ORCID iD and that it is validated in Editorial Manager

 We are very grateful to the editorial team for drawing our attention to this important requirement. We have now added the ORCID iD of the corresponding author— https://orcid.org/0000-0002-2247-5934 and other co-authors in the editorial manager and also on the title page of the manuscript

5. Please ensure that you include your title page within your main document. Yes, we have now added the title page to the main document.

Reviewers Comments: Reviewer #1

Introduction

1. The section is well written and provides clear gap of understanding how the interactions of institutional and socio-cultural factors impact in women’s decision making. Authors have also clearly presented need for the study and their position ‘Andersen and Newman Behavioural theoretical model for health service utilization’. This is great, which we do not commonly find in published papers.

 We very much appreciate the kind compliments from the reviewer. To further put emphasis on the theoretical basis of the study, we have extensively revised the abstract of the manuscript to ensure that the results are presented in line with the postulations of the Andersen and Newman model. We have also rephrased the title of the manuscript to directly reflect the real need and content of the manuscript.

Study setting

2. Authors can visualise study setting in country map. This might well locate international readers while reading the paper.

. 

 We thank the reviewer for making this important suggestion. We agree that a map will enable international readers appreciate the study context better. For this reason, we have now added a map that situates the study in the national context. The map also details the study communities.

3. Authors can shorten this section by only bringing key stats around HDI, female literacy and local health system context (which they have done in second paragraph). You might want to bring local targets around expected pregnancies and births—the proportion of women expected to give birth each year, and stats on at least 1 ANC visit

 We have shortened the section as recommended by the reviewer. We provided statistics on the number of expectant mothers registered for ANC, and the percentage of women who attended at least one 1 ANC visit.

Study Design

4. Authors can just keep the first sentence about the type of the design, the rest in the paragraph can be omitted as it basically adds no more new information. We thank the reviewer for the detailed and critical comments. We have heeded the reviewer’s advice and shortened the section. We have now merged the study design with study population and sampling because the section would have been too short to stand alone. 

5. Study Population and Sampling: I have asked couple of questions below just to add in the quality of the methodological process implemented: Authors might want to reword it ‘study population, sampling and recruitment’. We have revised the subtitle: it now reads “Study design, sampling and recruitment”

6. Can authors clarify how many women were in the list received from the St. Joseph? And, which year or months the list was extracted and why? The list contained 115 women who deliver at home. The list was extracted in July 2019for the purpose of this study. We have included this explanation in the methods section

7. .What were the criteria to recruit the 7 husbands purposively? Were these the husbands of the women interviewed? Or other husbands? We have clarified these issues raised as follows: “Four of the men were husbands of the sampled women while the three were other men. We purposively wanted to appreciate the views of the husbands of the women interviewed and other men for the purposes of data triangulation on the subject matter”. 

8. How many total participants did the authors approach? Were there any refusals? Because we purposively selected the participants, there were no refusals, all those contacted (50) participated in the study.

9. Can authors present a table with key socio-demographic characteristics (age, sex, ethnicity, number of ANCs in the last pregnancy, number of pregnancy, no of babies, place of previous childbirth, years of experience for midwives etc) of the participants from each group? This will provide a much better sense to the readers. In line with the second reviewer’s comments, we have provided some socio-demographic information of the respondents in the methods section. We have provided socio-demographic data on the women interviewed as follows: “The ages of the women ranged between 18-40 years. Eighteen (78%) of the women did not have any formal education and five of them (22%) had primary education. All the participants in the study were native Dagabas”. These were the only variables we covered in the study since it was purely qualitative and the purpose was not to determine association between socio-demographic characteristics and responses.

10. Did authors provide any incentives to these participants? We did not provide any incentives to the research participants. We would have reported it in the methods if he had provided any incentives for them. We visited all participants at their various homes for the data collection, so the participants did not incure financial cost that needed to be reimbursed and we did not have resources to provide incentives for each participant.

11. How did authors figure out that data saturation was reached at 50 respondents? Adding a few sentences about your practical insights might help future researcher, as this is mostly not a straightforward decision in the field. When did authors realise this—while collecting data in the field, at analysis stage? We included the following in the methods section to address the reviewer’s comment. 

“Based on a preliminary review of the data from this 50 respondents, by reading the field notes and listening to the tapes, we realised that no additional information was added from further response—thus data saturation was reached with the 50 respondents”

12. Write one or two sentences on how you validated English for the interviews taken in the local dialect. We have included the following statement to explain how we validated the translation of the questionnaire and response between English and Dagaare. “We validated the English for interviews conducted in the local dialect by back translating the interview guide in the local dialect in English..”

13. .Can authors provide the semi-structured interviews as additional file?

 We have uploaded both the English and Dagaare (local language) versions of the interview guide as an additional file in response to the reviewer’s and editorial requests.

14. Can authors also provide the thematic framework and the initial coding structure they used as additional files? We thank the reviewer very much for seeking clarity here. We believe the use of the word “framework” has made our argument unclear to the reviewer. We are actually referring to the analytical procedure we followed as detailed in the text. We have replaced the word “framework” with “analytical procedure. For this reason, we do not have a stand-alone “analytical framework” to upload as an additional file.

15. While 4 research assistants were used in data collection, how many were involved in coding and analysis?

 We thank the reviewer for his/her eye for detail. As detailed out in our description of the contributions of the authors, and in the methods section of our revised manuscript, we have now explained that the data analysis was led by two of the authors (first and second authors) and their work was validated by the third author 

16. . As authors said the framework used helped them to develop themes deductively—Can authors discuss this to what extent they were led by it? Did authors come up with any participants data which can challenge in any way to the already established framework they deployed? We are grateful to the reviewer for asking these questions. However, as we have indicated above, there is no stand-alone framework that was adopted but rather the five-staged analytical procedure we followed, which we have provided detailed explanation in the text.

17. If the matrix authors created is not too bulky, they can include that as an additional file Thank you very much for asking. Please, we do not have a stand-alone matrix to upload as an additional file. All analytical information and themes have been included in the manuscript.

Results

18. . Overall, the quotes presented provide quite powerful picture of why women end up giving birth at home. Several quotes author presented show adequate thematic prevalence. It is so interesting to read that all women indeed want to come to hospital. But, the lack of trust in facility births (especially operations and assisted births), faithfulness towards husbands, poor quality of facility birth (privacy, inadequate staff) are key factors which sustain home births.

Some specific comments below: We thank the reviewer for his/her kind words about our results section. The reviewer’s comments also gave us great insight and inspiration on why women attend ANC but deliver at home. Based on these comments have revised the title of the paper to read: “Why do women attend antennal care but give birth at home? A qualitative study in a rural Ghanaian District”. We believe this title speaks more directly to our results as observed by the reviewer. 

19. Authors said “For this reason, any woman who is given any form of injection, forceps or vacuum to help bring out a baby is considered unfaithful—which is why she could not deliver naturally on her own”---did authors mean she could not decide to seek healthcare on her own?

 Please, we meant that the woman could not a have “spontaneous vaginal delivery”. We have included this explanation in the revised manuscript

20. In the first quotation, ‘breath presentation’—did the health worker mean ‘breach presentation’?

 We thank the reviewer for this great observation. Yes, we meant “breach presentation” and NOT “breath presentation”. We have done the correction according and highlighted it in the text.

21. Likewise, authors can simply write ‘Woman and her age’, no need to repeat delivered at home if authors have interviewed only those who delivered at home. If available, authors might want to specify primi or multi. We appreciate this valuable input from the reviewer. We have deleted the “delivered at home” and added the ages of the women. We have no data on whether the women were primi or multi parous. 

22. What did husbands say on the ‘sign of faithfulness’? any quotes from husband?? We thank the reviewer for the question. We did not find specific quotes from husbands on the theme of “faithfulness.

23. The quotation on the religious belief you provided does not fully establish gods forbid, it rather establishes that women have indeed gone to hospital but did not have intended outcomes. What was the reasons behind her fresh stillbirths—delay in receiving care? Poor quality of intrapartum care? How common is this theme in your study? It would be great to support this with some additional quotes. We thank the reviewer for this important comment. 

This theme on the “god forbidding hospital delivery” was indeed among the common themes. The reason for the stillbirths was precisely based on the perception that if the women defied the gods and delivered at the hospital, they will have bad outcomes. As requested by the reviewer, we have added quotes in this regard as follows:

“I had wanted to deliver in the hospital but I was told by my mother in-law that all her eleven children (11) including my husband were all born in the house, for the gods, they worship are stronger than the white man’s medicine”.

The quote has been highlighted in the text in track changes.

24. Do people live in joint family or nuclear family in the study setting? How common is husbands’ accompaniment to their wives during ANC check-up? We thank the reviewer for the question. People generally live more in an extended family system in the study setting. We did not ask for information on husbands’ involvement in the maternal health care of their wives. However, available statistics suggest that male involvement in accompanying their wives for ANC is very low. For example, the annual report of the Jirapa Municipal Health directorate (2019), only 96 husbands accompanied their wives to ANC clinics.

Discussion

25. It is a great discussion. Yet, I would recommend authors to shorten it and discuss only those unique findings---interaction of religious beliefs and hospital births as sings of infidelity to their husbands and gods; women’s autonomy and the complex family context; lack of trust and poor satisfaction with health facility births; and the policy factors such as free childbirth policy. Authors might want to check my papers if it supports them in discussion in anyway We are grateful to the reviewer for these kind comments. We also thank him/her for referring us to his/her great work. We have cited some relevant aspects of the work, and we have appropriately cited it to support our discussion, which we believe has added value to our work. Here is the link to the paper we cited: https://journals.plos.org/plosone/article?id=10.1371/journal.pone.0194328.

26. While discussing the poor satisfaction and trust with the quality of care in facility births, authors might want to refer to literatures related to ‘respectful maternity care’.

 We thank the reviewer for drawing our attention to this important concept of “respectful maternity care” (RMC). As recommended by the reviewer, we have reviewed the literature on RMC and cited following papers

https://bmcpregnancychildbirth.biomedcentral.com/articles/10.1186/s12884-020-03118-0 and 

https://bmcpregnancychildbirth.biomedcentral.com/track/pdf/10.1186/s12884-020-03135-z.pdf to support our discussion. These sections have been highlighted in track changes in the text.

27. To save some words, authors might want to omit this from their first paragraph in the discussion section—“These strongly held traditional beliefs could perhaps, be attributed to the low educational status of women and their husbands in the study context. Because of the low educational levels of women and husbands, the women are not capacitated to detect when labour sets in while their illiterate husbands do not even appreciate the importance of delivering in the hospital. These findings corroborate earlier studies in Pakinstan, Ethiopia and Guinea-Bissau, which reported that educational status was a major predisposing factor associated with health facility delivery. For instance, existing quantitative studies, (26– 29) reported that women without formal education were more likely to deliver at home compared to women with higher levels of formal education. These findings demonstrate the empowering effect of education on women because women with higher levels of education are more likely to have increased knowledge of the benefits of health facility delivery, increased socialisation to interact with formal services outside the home environment, familiarity with modern medical culture, and access to increased financial resources. At the same time, husbands with higher levels of education are more likely to facilitate their wives’ motivation to deliver at the health facility (30).”

 We thank the reviewer for this suggestion. In accordance with the reviewer advice, we have deleted a large part of the section but not the entire section because we are guided by our conceptual framework. In addition, after carefully reviewing the manuscript we were convinced that deleting the entire section will distort the logical follow of the manuscript, and will importantly, down play the influence of education on place of delivery in the study setting. Thus, we have maintained this “These traditional beliefs could be attributed to the low educational status of women and their husbands in the study context. Studies in Pakinstan, Ethiopia and Guinea-Bissau reported that educational status was a major predisposing factor associated with health facility delivery. For instance, existing quantitative studies, [22–24] reported that women without formal education were more likely to deliver at home compared to women with higher levels of formal education. These findings demonstrate the empowering effect of education on women because women with higher levels of education are more likely to have increased knowledge of the benefits of health facility delivery, increased socialisation to interact with formal services outside the home environment, familiarity with modern medical culture, and access to increased financial resources. At the same time, husbands with higher levels of education are more likely to facilitate their wives’ motivation to deliver at the health facility”.

Conclusion and Recommendations

28. Authors can save words by omitting “The paper found very interesting and policy relevant results, including faithfulness to husbands, religious beliefs, women’s autonomy, and lack of, and cost of transportation as the major factors that predisposed most women to deliver at home. Institutional levels factors such as rude, negligent and unfriendly behaviour from health professionals, inadequate midwives, lack of privacy and confidentiality and hidden costs, and fear of caesarean delivery were found to be crucial barriers for health facility delivery. Our findings have far reaching policy implications for increasing health facility delivery and reducing maternal and child mortalities in consonance with SDG three.” From the conclusion and recommendation section.

Thank you for the great work.

 We grateful for the recommendation on how to save words in the concluding aspect of our work. We have revised this section. However, we have not deleted the entire section as suggested by the reviewer because we believe that by first giving a point summary of the findings, it will help readers, especially policy-makers, who may not have time to read the entire paper, to quickly appreciate the content of the paper and hence the basis/rationale for the recommendations that have been made.

Finally, the reviewer’s comments have been very helpful, and we thank him/her very much for contributing to improving the quality of paper.

Reviewer #2:

1. The authors have done a qualitative study to understand some socio-cultural and institutional factors that inhibit women from giving birth to child in hospital. This is a well argued article and can be accepted with a revisions on the following: We thank the reviewer for appreciating our work. In the revised title of the manuscript, we have now specifically drawn readers attention to the concepts of socio-cultural and institutional factors which form the main focus of the content of the paper.:

2. The authors organised the findings part in three major sections according to the given theoretical framework. However, in some subsections of the findings not adequate explanation of data, i.e., no or least explanation of the quotes is given. Only presenting quotes do not suffice to make the data presentation up to the standard. I suggest authors to revise the whole findings part, draw the findings and explanations from the quotes presented. We have noted with many thanks the reviewer’s comments. We acknowledge that they are great ones. For this reason, we have taken his/her advice seriously, and revised all the results section accordingly. All the revisions in the results section have been highlighted in tracked changes.

3. A brief discussion of socio-demographic information of the mothers interviewed need to be added in the methodology section. It can be added in the sub section 'study population and sample' We thank the reviewer very much for this important input. The first reviewer also pointed this out. Accordingly, we have presented the socio-demographic information of the mothers interviewed covering those variables we collected data on: age, educational level, and ethnicity in the methods section. We also included the ages of the women in their quotes.

4. The format of the quote, especially who gives the quote, needs to be uniformed. for example, for women's quote it is written (Woman who delivered at home) and for husbands' quote (IDI with husband). Patterns need to be uniformed.

 We are grateful to the reviewer for his/her eye for detail. We have now presented all the quotes in a uniform format. In all the quotes, from women and the husbands, we have now presented only the respondents and their ages, e.g. (woman, 18 year, husband, 45 years, etc).

---

## [Editor Report · Decision Letter 1]

15 Nov 2021

PONE-D-21-15389R1Why do women attend antennal care but give birth at home? A qualitative study in a rural Ghanaian DistrictPLOS ONE

Dear Dr. Kennedy Alatinga,

Thank you for submitting your manuscript to PLOS ONE. After careful consideration, we feel that it has merit but does not fully meet PLOS ONE’s publication criteria as it currently stands. Therefore, we invite you to submit a revised version of the manuscript that addresses the points raised during the review process.

You have revised your manuscript by appropriately addressing the comments and observations of the two reviewers. However, a few minor revisions are needed before the revised manuscript can be accepted for publication. These are stated as follows:

1. Please have another thorough editing for English. Please remember that the word "data" is a plural word (the singular is "datum"). Therefore, please state "data were" wherever you have stated "data is". In the Background section of the Abstract, write the letter "three" with a capital T (Sustainable Development Goal Three). Further, in the Results section (Traditional practices - home birth, a sign of faithfulness to husband), lines 7-8: you wrote "-- which is why she could not spontaneous vaginal delivery". Please correct this part of the sentence as "-- which is why she could not have a spontaneous vaginal delivery".

2. In the Introduction section, first line: Please update the statistics on global number of pregnancy related deaths. According to WHO, "About 295 000 women died during and following pregnancy and childbirth in 2017". (https://www.who.int/news-room/fact-sheets/detail/maternal-mortality).

3. In the Discussion section: Please cite a reference to the paper by Paudel et al.

4. In response to Comment 7 of Reviewer#1 you have stated that "“Four of the men were husbands of the sampled women while the three were other men", but in the text "Study design, population, sampling and recruitment", lines 11-12 you have written that: "Two of the men were husbands of the sampled women who delivered at home while the five were not." Please correct the this inconsistency.

5. In response Comment 14 of Reviewer#1, you have stated: "We believe the use of the word “framework” has made our argument unclear to the reviewer. We are actually referring to the analytical procedure we followed as detailed in the text. We have replaced the word “framework” with “analytical procedure". But you have still used the words "Theoretical Framework" in the text.

6. In response to Editorial Comment 3, if possible, please upload a copy of the transcript of the interviews as a supporting document.

We look forward to receiving your revised manuscript.

Kind regards,

Gouranga Lal Dasvarma, PhD

Academic Editor

PLOS ONE
---

## [Author Response · Author response to Decision Letter 1]

24 Nov 2021

1. We thank the review team for drawing our attention to this important error. We have now addressed and changed all the statements that indicated, “data is/was” to “data were”. These changes are marked in tracked changes in the abstract and on page 7 of the manuscript. The manuscript has also now been thoroughly edited for English by an expert.

a). We thank the review team for their kind support. We have corrected, “The Sustainable Development Goal three”, to now read “Sustainable Development Goal Three”, in the abstract, introduction, and the conclusion recommendations sections of manuscript on pages 2, 3 and 20 respectively. 

b). Please, the sentence has now been corrected to read, “... which is why she could not have a spontaneous vaginal delivery”. 

2. We are grateful to the reviewers for drawing our attention to these important updates. We have accordingly updated the statistics in the first line of the introduction as follows, “..the World Health Organization estimates that about 295 000 women died due to causes related to pregnancy and childbirth in 2017”.

Reference #1 below now replaces reference #2 also below as reference #1 in the reference list in the main manuscript.

1). World Health Organization. Maternal mortality [Internet]. 2019 [cited 2021 Nov 16]. Available from: https://www.who.int/news-room/fact-sheets/detail/maternal-mortality

2). World Health Organization. WHO | Proportion of birth attended by a skilled health worker [Internet]. WHO. World Health Organization; 2008 [cited 2021 Mar 7]. Available from: https://www.who.int/reproductivehealth/publications/maternal_perinatal_health/2008_skilled_attendants/en/

3. We thank the review team for drawing our attention to this issue. We have duly cited Paudel et al’s paper. The reference is captured as [19] on page 16 of manuscript.

4. Thank you for drawing our attention to this inconsistency. The response to Comment 7 of reviewer #1 was an error. The right statement is what is contained in the main manuscript on page 7. That is, “Two of the men were husbands of the sampled women while the five were other men.”

5. Thank you very much for this great input. We have now carefully read through the manuscript and replaced the word “framework” with “analytical procedure”, as marked in track changes on page 8 of the manuscript. Again, on page 4, we have replaced the sub-title “Theoretical framework of the study” with “Theoretical model of the study”, for consistency with the Andersen and Newman Behavioural theoretical model for health service utilization.

6. As we indicated in our previous response, this study is based on a dataset of 50 qualitative interview transcripts. However, we did not seek ethical permission from the participants or the ethics committee, to use data for anything else, other than the specific purposes of this study. For this reason, we do not have the explicit permission for data sharing, re-analysis or future studies. It would therefore, be inappropriate and unethical to make them available in the public domain. Furthermore, data cannot be shared publicly because the individual transcripts contain very sensitive and identifying personal information from the participants and we did not obtain consent from the participants nor the ethics committee to upload such information for public sharing. Thus, making the transcripts publicly available, will only lead to ethical violations.

---

## [Editor Report · Decision Letter 2]

1 Dec 2021

Why do women attend antennal care but give birth at home? A qualitative study in a rural Ghanaian District

PONE-D-21-15389R2

Dear Dr. Alatinga,

We’re pleased to inform you that your manuscript has been judged scientifically suitable for publication and will be formally accepted for publication once it meets all outstanding technical requirements.

Kind regards,

Gouranga Lal Dasvarma, PhD

Academic Editor

PLOS ONE

Additional Editor Comments (optional):

Thank you for addressing all the comments.
---

## [Editor Report · Acceptance letter]

7 Dec 2021

PONE-D-21-15389R2 

Why do women attend antenatal care but give birth at home? A qualitative study in a rural Ghanaian District 

Dear Dr. Alatinga:

I'm pleased to inform you that your manuscript has been deemed suitable for publication in PLOS ONE. Congratulations! Your manuscript is now with our production department. 

Kind regards, 

on behalf of

Dr. Gouranga Lal Dasvarma 

Academic Editor

PLOS ONE